# Application of Free Satellite Imagery to Map Ecosystem Services in Ungwana Bay, Kenya

Daina Mathai [1,2,3,*], Sónia Cristina [1] and Margaret Awuor Owuor [4,5,6]

1   CIMA—Centre for Marine and Environmental Research/ARNET-Aquatic Research Network, Universidade do Algarve, Campus de Gambelas, 8005-139 Faro, Portugal
2   Department of Biological, Geological and Environmental Sciences, University of Bologna, 48123 Ravenna, Italy
3   Center for Coastal and Ocean Mapping, University of New Hampshire, Durham, NH 03824, USA
4   Wyss Academy for Nature at the University of Bern, 3011 Bern, Switzerland
5   Institute of Ecology and Evolution, University of Bern, 3011 Bern, Switzerland
6   School of Environment Water and Natural Resources, South-Eastern Kenya University, Kitui P.O. Box 170-90200, Kenya
*   Correspondence: daina.mathai@unh.edu

**Abstract:** A major obstacle to mapping Ecosystem Services (ES) and the application of the ES concept has been the inadequacy of data at the landscape level necessary for their quantification. This study takes advantage of free satellite imagery to map and provide relevant information regarding ES and contribute to the sustainable management of natural resources in developing countries. The aim is to assess the flow of ES in mangrove ecosystem of Ungwana Bay, located on the northern coast of Kenya, by adopting the Land Use Land Cover (LULC) matrix approach. This study characterized LULC classes present in the study area, identified the most important ES, and collected data on expert opinions via a survey on ES flow supplied by the mangrove ecosystem. A qualitative and quantitative analysis of the expert scoring produced a LULC matrix which, when integrated with the LULC maps, showed the spatial distribution of ES flow. The assessment indicates very high flow (5.0) for the regulating and supporting services, high flow (4.0) for the cultural services, and medium flow (3.0) for the provisioning services as supplied by mangroves. In addition, the analysis indicates there are sixteen major ES supplied by the mangrove ecosystem of Ungwana bay as of the year 2021. This study highlights the importance of mangroves as a coastal ecosystem and how the visualization of the spatial distribution of ES flow using maps can be useful in informing natural resource management. In addition, the study shows the possibilities of using freely accessible satellite imagery and software to bolster the ES assessment studies lacking in developing countries.

**Keywords:** mapping; ecosystem service flow; satellite remote sensing; mangrove; Kenya





## 1. Introduction

Coastal ecosystems are some of the most productive systems on earth [1], being the planet's life-support systems for the human species and all other forms of life [2]. Among these systems, are the mangrove ecosystems that provide a wide range of goods and services to both nature and society being viewed at a local, national, and global scale [3]. How the ecosystem's structure and function, with the combination of other inputs, contribute to human well-being is what is referred to as Ecosystem Services (ES) [4]. However, the dual trends of local and global population growth and the increase in consumption rates have led to the increased demand for coastal ES. Between 1985 and 2005, the world lost about 35% of mangrove forests; they are declining faster than tropical forests and coral reefs [3]. Land Use Land Cover (LULC) change and global climate change have been cited as major threats to mangrove forests [5].

Mangroves capture and preserve significant amounts of carbon; they have the highest carbon pools of any forest type [5]. Mangroves and other carbon-rich coastal ecosystems such as seagrass and tidal marshes commonly referred to as "blue carbon ecosystems" can counterbalance anthropogenic greenhouse gases, playing a key role in mitigation and resiliency of climate change-related effects [1,5].

Intensive LULC change in the coastal areas to meet the demand of the rapidly growing population has placed coastal ecosystems under threat. This threat is altering the flow of ES, leading to irreversible damage and a decline in the supply of services to both society and environment [6]. The degradation of mangroves also leads to the gradual release of large amounts of carbon back into the environment from their high-carbon pools [6]. This rapid change in the last second half of the 20th century has led to a significant barrier to achieving Sustainable Development Goals (SDGs) and fear that consequences will overflow into the next coming centuries [7].

For these reasons, ES has gained much popularity in research and has grown tremendously, including being applied as a conceptual framework for projects such as the Millennium Ecosystem Assessment [8]. The ES concept aims to understand the ecosystem components and processes and their biophysical potential to provide ES to society. As such, ES assessment has become a powerful tool in scientific literature and mapping techniques, integrating the complex nature of ES into environmental management and decision-making [9]. Thus, it has been identified as a key element in supporting adaptive management [9–13].

Satellite Remote Sensing (SRS), coupled with the advancement of GIS technology [14], has become an important tool in ES assessment by producing ES spatial maps for either ES supply or demand [10]. SRS provides higher spatial, spectral, and temporal resolutions with the availability of historical data, making it possible to map and monitor coastal ecosystems that occur in tide-inundated and inaccessible regions [15] complementing in situ data as well as acting as a reconnaissance tool [16,17]. The SRS data on "free access policy" from Landsat and Sentinel satellite missions play an important role in monitoring natural resources, especially in developing countries where funding for the acquisition of SRS data is limited [18,19].

This work aims to assess the flow of ES from the mangrove forests of Ungwana bay on the north coast of Kenya. It adopts the LULC matrix developed by Burkhard et al. (2009) [10] to identify ES, characterize the LULC classes present, and provide visualization of the flow of ES in Ungwana bay using spatial distribution maps. This approach has been applied successfully in studies by Palomo et al. [9], Burkhard et al. [7], Owuor et al. [12], and Müller et al. [13] by integrating expert opinions to land cover maps to understand the supply of ES in a specific landscape. The methodology adopted in this study builds on the work of Burkhard et al. [10] by applying the LULC matrix at a local scale in developing countries and further building on the work of Kirui et al. [16], mapping the mangroves using freely accessible medium resolution satellite imagery.

## 2. Materials and Methods

### 2.1. Study Area

The study was carried out along the lagoons and tidal flats of Ungwana bay, northern banks of Kenya (NBK) stretching between 30.881°E, 2.407°S and 40.699°E, 3.280°S. Ungwana bay is a wide shallow embayment in front of River Sabaki on the south and River Tana on the north, separated from Malindi bay by the Ngomeni peninsular (Figure 1).

The study area is categorized as block 8 in Kenya's National Mangrove Plan (NMP) zonation [20]. The mangrove forest in the study area covers a total area of 4240 ha comprising riverine, fringing, and creek mangroves. They have a stocking density of 2015.2 stems per hectare and a volume of 187.5 m$^3$. The mean height is recorded at 6 m with a diameter of 8.1 cm, making it 34% merchantable [20]. In the Area of Interest (AOI), at least 8 out of 9 species of mangrove found on the Kenyan coast are present with *Rhizophora mucronata* and *Avicennia marina* dominating. Other aquatic flora present in and around the mangrove

ecosystem include seagrasses, algae and fungi, grasses and sedges, and terrestrial plants such as neem, palms, and pines [21].

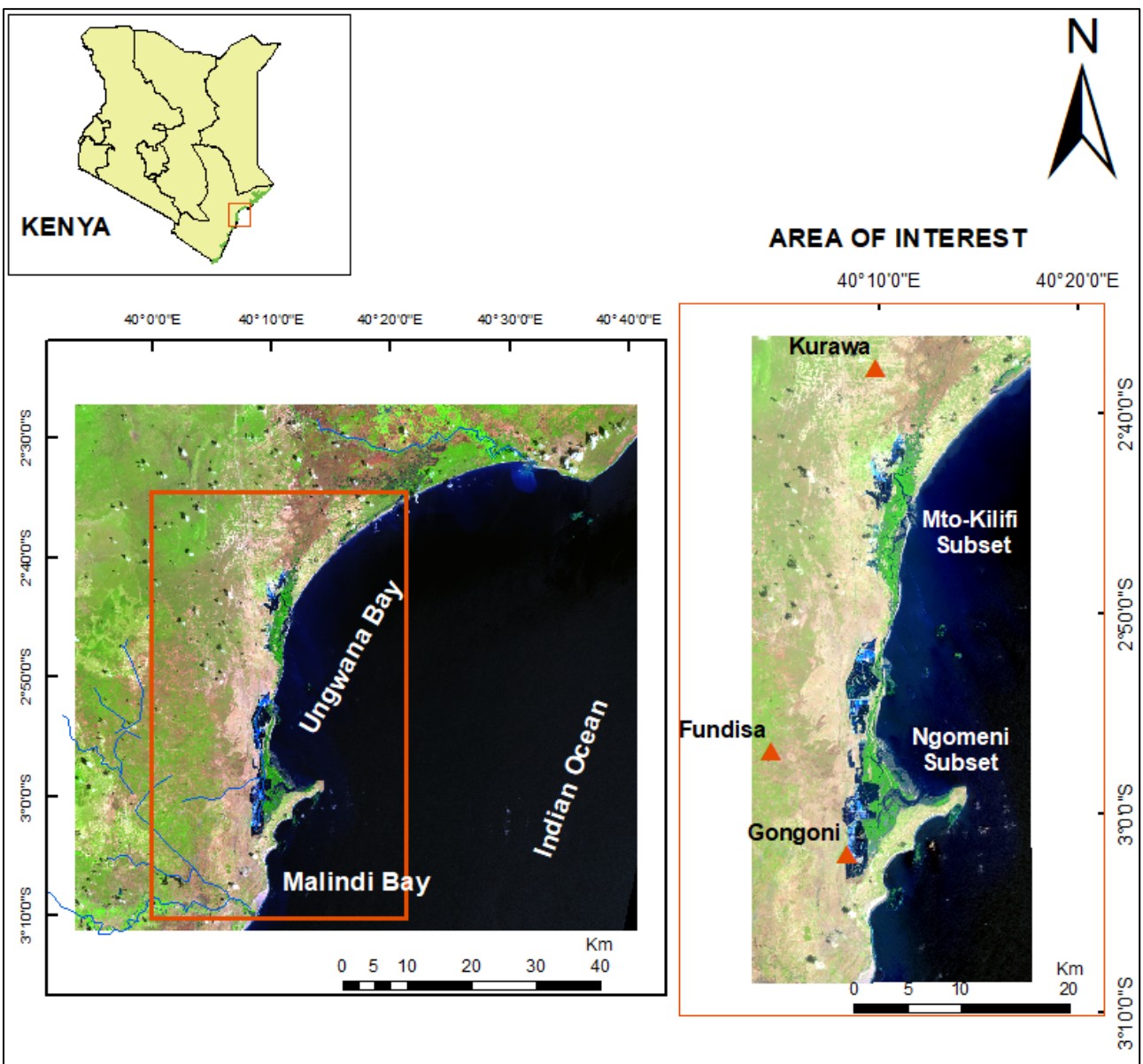

**Figure 1.** The geographic location of the study area and the area of interest (red triangles) on the North coast of Kenya (Source: Landsat-8 image courtesy of the U.S. Geological Survey).

Mangroves in the study area occur in patches; hence, this study will focus on the mangroves of the lower part of Ungwana bay, formed of the two subsets of Ngomeni and Mto Kilifi (Figure 1). This choice was also made to maneuver the cloud cover limitation on satellite images. In addition, sub-setting an AOI is commonly applied during LULC classification analysis as it improves accuracy by reducing the number of landcover types and spectral variations [19].

### 2.2. Land Use Land Cover Matrix Approach

This study adopted the LULC matrix approach by Burkhard et al. [10]. This approach relies on the integration of expert opinion with LULC maps in a quantitative and qualitative assessment, indicating the distribution of ES flow. The steps followed for this study include:

(1) identification of mangrove ES; (2) adopting the ES matrix approach to score the ES flow of each land cover type; (3) integrating the ES flow scores with LULC maps.

### 2.2.1. Identification of Mangrove Ecosystem Services

The most important mangrove ES were identified following the outline in the Millennium ecosystem assessment report [2] and narrowed down using literature of the local mangrove studies [12,16,20,22–24]. The generated list of ES (Table 1) was then used in a survey to guide the respondents in identifying ES found in the mangrove ecosystem of the study area. An additional question was posed to state any other mangrove ES not listed.

The survey targeted mangrove "experts" which included those in academia, government, and non-government sectors working with mangroves along the Kenyan coast and community members living adjacent to mangrove forests in the study area and/or those involved in mangrove activities. Data were collected using a structured questionnaire by research assistants in the field for the community groups. An online remote platform (https://ec.europa.eu/eusurvey/home/welcome; accessed on 15 December 2020) was also used to target those in the academia, government, and non-government sector due to their ability to access the internet. The administration of the questionnaire was equally distributed across the two groups targeting at least 100 sample size employing random sampling technique. The research assistants provided clear explanations of concepts in the survey, especially to the community groups in the language they easily understood.

### 2.2.2. Quantification of Ecosystem Service Flow

To quantify the flow of ES, the LULC matrix developed by Burkhard et al. [10] was adopted. The matrix questions (Supplementary material) were scored by respondents as follows: 0 = no flow, 1 = very low flow, 2 = low flow, 3 = medium flow, 4 = high flow, and 5 = very high flow of each ES in the specific LULC class. A total of 9 LULC classes were identified but only 6 classes were later assessed to concur with the characterized classes from the satellite imagery. Measures of means of the flows were calculated in Microsoft Excel, with each ES across the 6 LULC classes within the 0 to 5 scale mentioned above indicating the level of flow of each ES. The colour scheme was modified from Burkhard et al. [10] in creating the matrix table where 0 = rosy colour (no flow), 1 = grey-green (very low flow), 2 = light green (low flow), 3 = peridot green (medium flow), 4 = tarragon green (high flow), and 5 = dark green (very high flow).

### 2.2.3. Integration of LULC Map and Matrix Scores

This study applied satellite imagery obtained from Sentinel-2A (4 February 2021) and Sentinel-2B (19 February 2021) multi-spectral instrument (MSI) payload used to characterize the LULC classes present in the study area due to its high spatial resolution (10 m). Sentinel-2 satellite images were obtained from the European Space Agency (ESA) Copernicus Open Access Hub website [25].

Using the ESA Sentinel Application Platform (SNAP-v7.0.3), the selected images were pre-processed by increasing the resolution, geometrically resampling the Sentinel-2 data products at 10 m (Band 2) spatial resolution. The resampled images were then reprojected using the WGS84/UTM 37S, in the AOI.

Classification of LULC classes in the AOI was conducted using the Semi-automatic Classification plugin (SCP) in QGIS (v.3.16.5). Supervised classification was adopted following four main steps suggested by Leroux et al. [26]: (1) creation of training samples; (2) definition and analysis of spectral signatures of the training samples; (3) land use classification; (4) post-processing work.

The flow scores from the matrix table for each category of ES (provisioning, regulating, supporting, and cultural services) were then joined with the attribute table of the LULC maps, integrating the two forms of data to display the spatial distribution of ES flow.

**Table 1.** A description of the surveyed ES in Ungwana bay, Kenya was adapted with permission from Owuor et al. [12].

| Ecosystem Service | Definition Ecosystem Service Used in This Study |
|---|---|
| Provisioning services | |
| Wood products | Include mangrove products used in construction such as timber, poles, and fishing gear. |
| Fuel | Include mangrove products used as a source of fuel such as firewood and charcoal. |
| Freshwater | Water for domestic use. |
| Fisheries | All forms of seafood including aquaculture and fishing baits harvested, as well as the role of the mangrove ecosystem as a nursery and spawning ground for fish. |
| Wild food and honey | Foods harvested in the mangrove ecosystem such as berries, vegetables, and honey. |
| Local employment (This ES was introduced by the respondents during the survey) | Sources of income associated with the mangrove ecosystem of Ungwana bay. |
| Regulating services | |
| Carbon sequestration | Below and above ground carbon storage in the mangrove ecosystem and its role in regulating local and global climate. |
| Water purification | The ability of the mangrove ecosystem to filter and purify water against sediments, debris, and all forms of pollution. |
| Shoreline protection | Protection of the shoreline against erosion and effects of river/estuary floods and/or sea-level rise. |
| Preservation of biodiversity | The role of the mangrove ecosystem is to preserve all kinds of biodiversity in Kingdom Plantae and Animalia. |
| Supporting services | |
| Sediment trapping | The role that mangrove's root structure plays in trapping sediments and filtering from rivers, run-off, and sea backwash. |
| Nutrient cycling | The role that the mangrove ecosystem plays in providing nutrients from its rich organic matter within and without other adjacent ecosystems. |
| Cultural services | |
| Recreation and Tourism | Any activities associated with the mangrove ecosystem are enjoyed in Ungwana bay such as canoe riding and bird watching by both local and international tourists. |
| Cultural heritage | Any cultural importance or benefit attached to the mangrove ecosystem of Ungwana bay such as shrines. |
| Education and Research | Formal and informal education is derived from the mangrove ecosystem of Ungwana bay by the locals and those in academia. |

*2.3. Data Analysis*

Descriptive data analysis was conducted using MS Excel and R software to assess their significance. In this statistical analysis case, we sought to find out the significance of the results ($p$ value), considering the null hypothesis ($H_O$) that there is no relationship between the ES being investigated and the LULC classes in which it is supplied, while the alternative hypothesis (Hi) that there is a significant correlation between the ES being investigated and the LULC classes it is supplied. In the analysis, only LULC classes with high scores ($\geq 2$) were identified and used in the one-way ANOVA analysis. This was to reduce the chances of outliers that would affect the sample variance, decreasing the F statistic for

the ANOVA and lowering the chance of rejecting the null hypothesis. Using R script, a one-way ANOVA statistic was applied because only one factor was being accounted for, with ES being the independent variable and the LULC classes supplying ES as dependent variables. However, following this criterion, the water purification ES presented a case of a small sample size, i.e., only the mangrove LULC class had a high score of $\geq 2$. Therefore, we conducted a chi-square test for the water purification ES to calculate its *p*-value.

## 3. Results

### 3.1. Land Use Land Cover Classes

The analysis of LULC classification of Mto-Kilifi (Figure 2) and Ngomeni (Figure 3) data resulted in six LULC classes, namely, waterbodies (the ocean, creeks, flooded salt ponds), mangrove forests, sandflats (the mudflats, intertidal areas, sand ridges, and the beach area), settlements (the built-up area), other vegetation types (all other vegetation that is not mangroves), and bare areas. From the legend inside the figures, the area in hectares (Ha) of each class was calculated as of February 2021, with mangrove forests occupying 1680.39 Ha in the Mto-Kilifi and 1740.19 Ha in the Ngomeni.

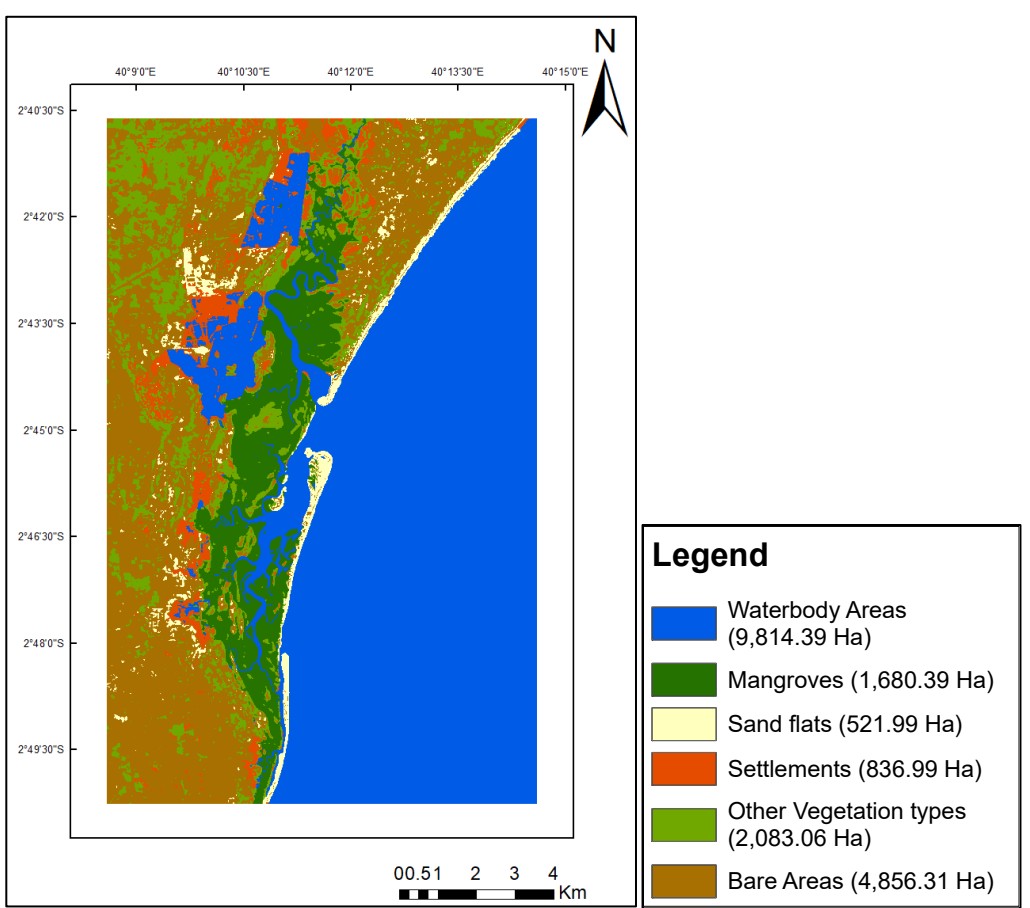

**Figure 2.** LULC classes and the area (Ha) of each class present in Mto-Kilifi (Ungwana bay, Kenya) were processed using Sentinel-2 satellite image sensed in February 2021.

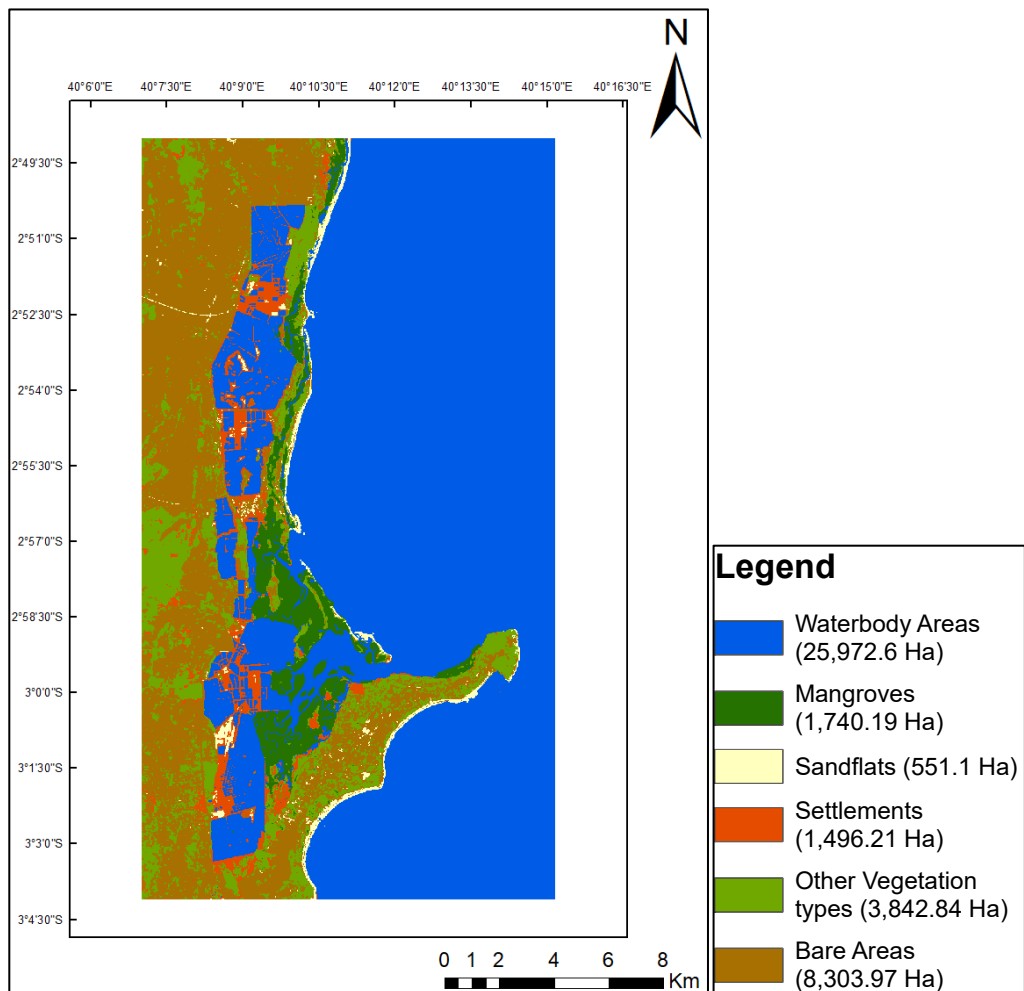

**Figure 3.** LULC classes and the area (Ha) of each class present in Ngomeni (Ungwana bay, Kenya) were processed using Sentinel-2 satellite image sensed in February 2021.

### 3.2. Mangrove Ecosystem Services

Out of the 100 questionnaires disseminated, an 80% response rate was achieved from the survey (Supplementary material). On compiling the data to identify ES present in the study area, sixteen (16) ES were identified as the most important ES provided by the mangrove ecosystem of Ungwana bay (Figure 4).

Two-thirds (68%) of the respondents were of the opinion that the mangroves of Ungwana bay provide the 16 listed ecosystem services (Figure 4) with the exceptions of freshwater, water purification, and local employment. The response rate in associating freshwater with the mangrove ecosystem was very low (10%). In response to the question, "list any other ES not listed?", 37.5% of the respondents identified local employment as an additional benefit derived from the mangrove forest.

One hundred percent of the respondents, which equates to 80 responses, agreed that the mangrove forest of Ungwana bay played a key role in sediment trapping. As many as 62.5% of the respondents identified nutrient cycling, cultural heritage, and education research as an important ES, while recreation and tourism scored highly, with 92.5% of the respondents associating it with the mangrove ecosystem (Figure 4). Further statistical testing revealed there was a significant difference ($p < 0.001$) in the identification of the ES supplied by the mangrove ecosystem of Ungwana bay.

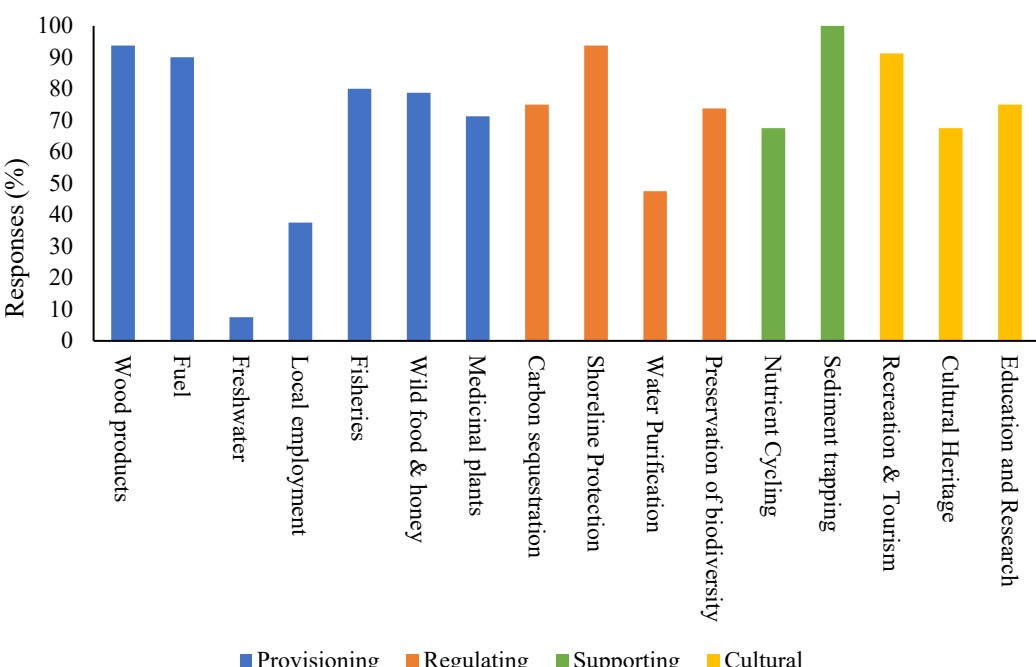

**Figure 4.** The responses of each ES were assessed from the survey administered to mangrove experts between January and April 2021 in Ungwana bay, Kenya.

*3.3. Quantification of Ecosystem Service Flow*

In this section, the respondents scored each ES provided from each LULC class identified. The means of all the scores were calculated and are presented in the matrix table below (Table 2). The scores were made on a scale of 0–5 where 0 = no flow, 1 = very low flow, 2 = low flow, 3 = medium flow, 4 = high flow, and 5 = very high flow.

**Table 2.** Matrix for the assessment of the different LULC classes' capacities to supply selected ES in Ungwana bay, Kenya. The assessment scale 0 = rosy colour (no flow), 1 = grey-green (very low flow), 2 = light green (low flow), 3 = peridot green (medium flow), 4 = tarragon green (high flow), and 5 = dark green (very high flow).

| LULC Classes | Provisioning ES | Wood products | Fuel | Wild food and honey | Local employment | Medicinal Plants | Fisheries | Freshwater | Regulating ES | Carbon sequestration | Shoreline protection | Water Purification | Preservation biodiversity | Supporting ES | Nutrient Cycling | Sediment trapping | Cultural ES | Recreation and Tourism | Cultural Heritage | Education and Research |
|---|---|---|---|---|---|---|---|---|---|---|---|---|---|---|---|---|---|---|---|---|
| Mangroves [a] | | 4 | 4 | 3 | 3 | 3 | 5 | 2 | | 5 | 5 | 3 | 5 | | 4 | 5 | | 4 | 3 | 4 |
| Other Vegetation types [b] | | 3 | 4 | 2 | 2 | 3 | 1 | 2 | | 3 | 2 | 1 | 3 | | 2 | 3 | | 3 | 2 | 3 |
| Water bodies [c] | | 0 | 0 | 0 | 1 | 0 | 3 | 1 | | 1 | 0 | 1 | 2 | | 2 | 1 | | 2 | 1 | 2 |
| Sandflats [d] | | 0 | 0 | 0 | 1 | 0 | 1 | 0 | | 0 | 1 | 0 | 1 | | 1 | 2 | | 2 | 1 | 2 |
| Settlements [e] | | 1 | 1 | 1 | 3 | 1 | 1 | 1 | | 0 | 0 | 0 | 1 | | 0 | 1 | | 1 | 2 | 2 |
| Bare Areas [f] | | 0 | 0 | 0 | 0 | 0 | 0 | 0 | | 0 | 0 | 0 | 1 | | 0 | 0 | | 0 | 1 | 1 |

[a] Mangroves comprise of the mangrove forest. [b] Other vegetation types include all other vegetation that are not mangroves. [c] Waterbodies comprise of the ocean, creeks, and the flooded salt ponds. [d] Sandflats include the intertidal mudflats, sand ridges, and the beach area. [e] Settlements comprise of built areas. [f] Bare areas are all open non-vegetated saline areas.

3.3.1. Flow of the Provisioning Ecosystem Services

Mangroves and other vegetation types were the major sources of wood products and fuel in Ungwana bay. Fuel (80%), which comprises firewood and charcoal, and medicinal plants (60%) showed similar flows from both mangroves and other vegetation types. Mangroves scored highest (100%) for the provision of fisheries as an ES, while the same system scored lowest (20%) for the provision of freshwater (20%). Further, the one-way ANOVA test on whether there was a relationship between the provisioning services and the LULC classes with high scores (Table 3) revealed that there was a significant correlation ($p = 0.703$) with many of the provisioning services being provided by the LULC classes. The results of freshwater and medicinal plant services were not significant ($p = 0.373$) (Table 3).

**Table 3.** Summary statistics of the one-way ANOVA-test for the assessment of flow provisioning ES from the LULC with high scores: n = 80 for Ungwana bay, Kenya.

| Provisioning ES | LULC Class with a Score of $\geq 2$ | Mean Scores of Flows | F Values | *p* Values |
|---|---|---|---|---|
| Wood products | Mangroves | 4.0 | 50.05 | $p < 0.001$ |
| | Other Vegetation types | 3.0 | | |
| Fuel | Mangroves | 4.0 | 9.42 | $p < 0.001$ |
| | Other Vegetation types | 4.0 | | |
| Wild food and honey | Mangrove | 3.0 | 13.01 | $p < 0.001$ |
| | Other Vegetation types | 2.0 | | |
| Local employment | Mangroves | 3.0 | 12.51 | $p < 0.001$ |
| | Other Vegetation types | 2.0 | | |
| | Settlements | 3.0 | | |
| | Sandflats | 2.0 | | |
| Medicinal plants | Mangroves | 3.0 | 0.8 | $p = 0.373$ |
| | Other Vegetation types | 3.0 | | |
| Fisheries | Mangroves | 5.0 | 53.77 | $p < 0.001$ |
| | Water bodies | 3.0 | | |
| Freshwater | Mangroves | 2.0 | 0.15 | $p = 0.703$ |
| | Other Vegetation types | 2.0 | | |

3.3.2. Flow of Regulating Ecosystem Services

Mangroves were scored as providing very high flow (100%) for carbon sequestration, shoreline protection, and the preservation of biodiversity. Other vegetation types are also identified as important for carbon sequestration (60%) and the preservation of biodiversity (60%).

A one-way ANOVA test on whether there was a relationship between the regulating services and the LULC classes with high scores (Table 4) showed that the scoring of the flow of the three regulating services, i.e., as provided by mangroves, other vegetation types, and sandflats, was significant ($p < 0.001$). In the outlier case of water purification which had a small sample size, the results also indicated a significant correlation between the scoring of the water purification service as supplied by the mangrove LULC class (Table 4).

**Table 4.** Summary statistics of the one-way ANOVA-test for the assessment of flow regulating ES from the LULC with high scores: n = 80 for Ungwana bay, Kenya.

| Regulating ES | LULC Class with a Score $\geq 2$ | Mean Scores of Flows | F Values | p Values |
|---|---|---|---|---|
| Carbon sequestration | Mangroves | 5.0 | 120.31 | $p < 0.001$ |
| | Other Vegetation types | 3.0 | | |
| Shoreline protection | Mangroves | 5.0 | 113.15 | $p < 0.001$ |
| | Other vegetation | 2.0 | | |
| | Sand flats | 2.0 | | |
| Water purification * | Mangrove | 3.0 | — | $p < 0.001$ |
| Preservation of biodiversity | Mangroves | 5.0 | 56.18 | $p < 0.001$ |
| | Other Vegetation types | 3.0 | | |
| | Water bodies | 2.0 | | |

* The water purification service scored very poorly in other LULC classes therefore a chi-square test was conducted to find the *p* value. This applied for the F statistic as well as the service had a high score in Mangroves LULC only.

### 3.3.3. Flow Supporting Ecosystem Services

The results, as shown in (Table 5) below, indicate that mangroves scored highest in their role in sediment trapping (100%) and nutrient cycling (80%). Other vegetation types had a low flow (40%) to nutrient cycling and a medium flow (60%) to sediment trapping.

**Table 5.** Summary statistics of the one-way ANOVA-test for the assessment of flow supporting ES from the LULC with high scores: n = 80 for Ungwana bay, Kenya.

| Supporting ES | LULC Class with a Score $\geq 2$ | Mean Scores of Flows | F Values | p Values |
|---|---|---|---|---|
| Nutrient cycling | Mangroves | 4.0 | 31.78 | $p < 0.001$ |
| | Other Vegetation types | 2.0 | | |
| | Water channels | 2.0 | | |
| Sediment trapping | Mangroves | 5.0 | 77.26 | $p < 0.001$ |
| | Other Vegetation types | 3.0 | | |
| | Sand flats | 2.0 | | |

A one-way ANOVA test on whether there was a relationship between the supporting services and the LULC classes with high scores (Table 5) showed that the scores for the flow of the two supporting services assessed, from the LULC classes with high scores, were significant ($p < 0.001$).

### 3.3.4. Flow of Cultural Ecosystem Services

The results of the cultural ES assessment revealed that mangroves had a high flow for recreation and tourism and education and research and a medium flow for cultural heritage. The scoring of the flow of the cultural services across LULC classes with high scores was significant ($p < 0.001$). What is interesting about the data in this category is how recreation and tourism and education and research scored highly ($\geq 40\%$) across several LULC classes (Table 6).

**Table 6.** Summary statistics of the one-way ANOVA-test for the assessment of flow cultural ES from the LULC with high scores: n = 80 for Ungwana bay, Kenya.

| Cultural ES | LULC Class with a Score $\geq$ 2 | Mean Scores of Flows | F Values | *p* Values |
|---|---|---|---|---|
| Recreation and Tourism | Mangroves | 4.0 | 39.8 | $p < 0.001$ |
| | Other Vegetation types | 3.0 | | |
| | Sandflats | 4.0 | | |
| | Water bodies | 2.0 | | |
| Cultural heritage | Mangroves | 3.0 | 13.62 | $p < 0.001$ |
| | Settlements | 2.0 | | |
| Education and Research | Mangroves | 4.0 | 38.57 | $p < 0.001$ |
| | Other Vegetation types | 3.0 | | |
| | Water bodies | 2.0 | | |
| | Sandflats | 2.0 | | |

*3.4. Spatial Distribution of Ecosystem Services Flow*

The results of integrating the matrix table data (Table 2) and LULC class maps of Mto-Kilifi (Figure 2) and Ngomeni (Figure 3) are presented below, showing the spatial distribution of the four categories of ES. Figures 5 and 6 show the spatial distribution of the flow of the four categories of ES (provisioning, regulating, supporting, and cultural) of Mto-Kilifi and Ngomeni, respectively, as supplied by the mangrove ecosystem of Ungwana bay.

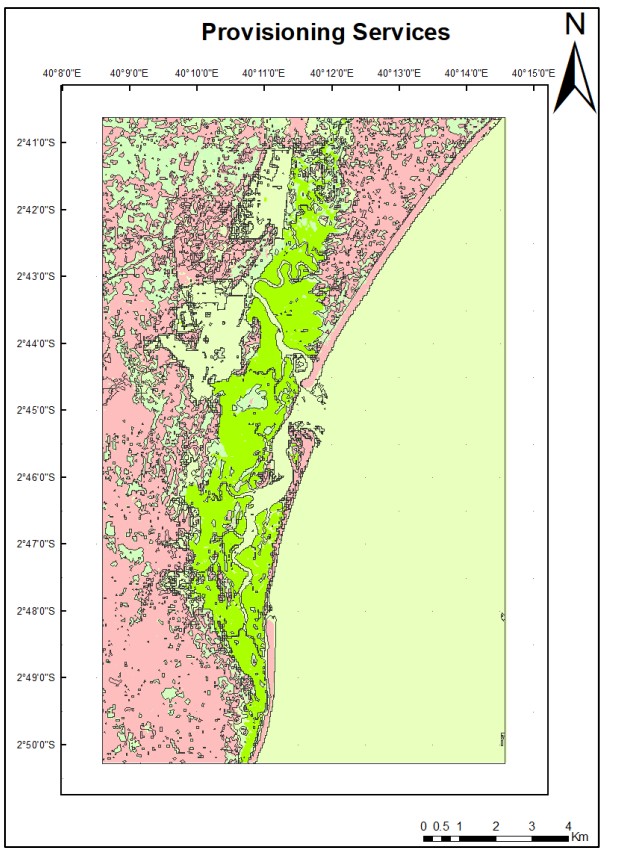 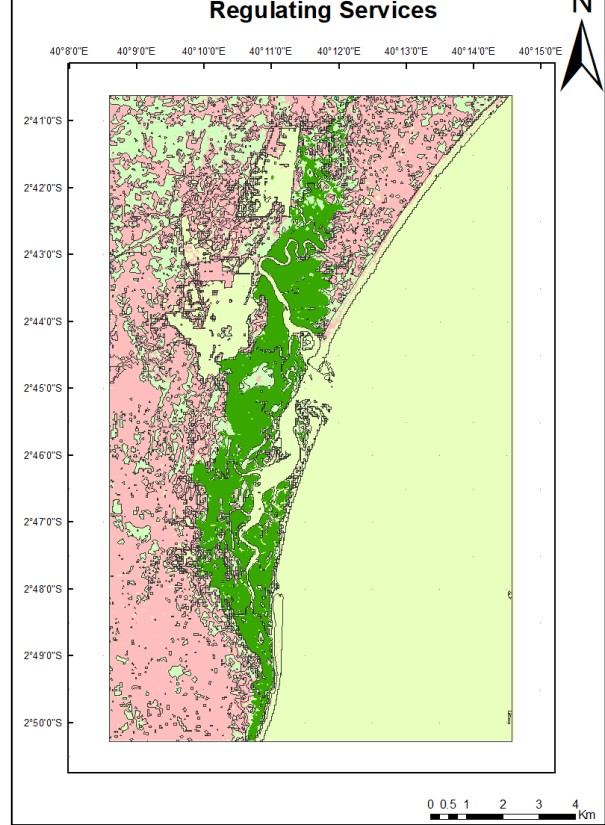

**Figure 5.** *Cont.*

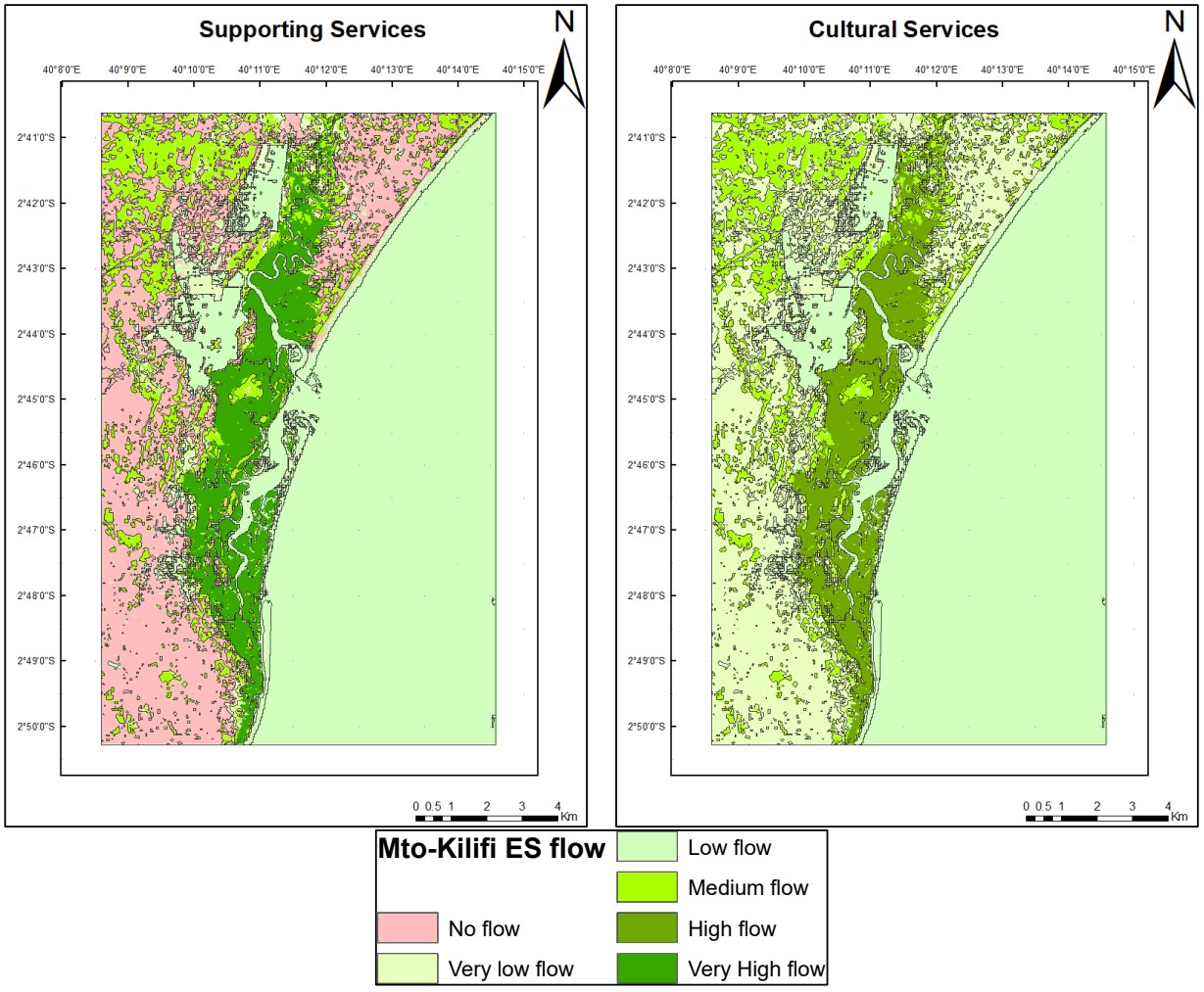

**Figure 5.** The spatial distribution of ES flow for provisioning, regulating, supporting, and cultural services provided in Mto-Kilifi (Ungwana bay, Kenya).

There is very high flow (5.0) of the regulating services (carbon sequestration, shoreline protection, water purification, preservation of biodiversity) and supporting services (sediment trapping, nutrient cycling) from the mangroves of Mto-Kilifi and Ngomeni. Further, the cultural services (recreation and tourism, cultural heritage, education, and research) indicate a high flow (4.0), while the provisioning services (wood products, fuel, fisheries, freshwater) indicate a medium flow (3.0) from the mangroves of Mto-Kilifi and Ngomeni. It should be noted that the Mto-Kilifi and Ngomeni subsets (Figure 1) were used to counter the cloud interference on the satellite imagery; otherwise, it is simply one location with similar environmental and demographic characteristics.

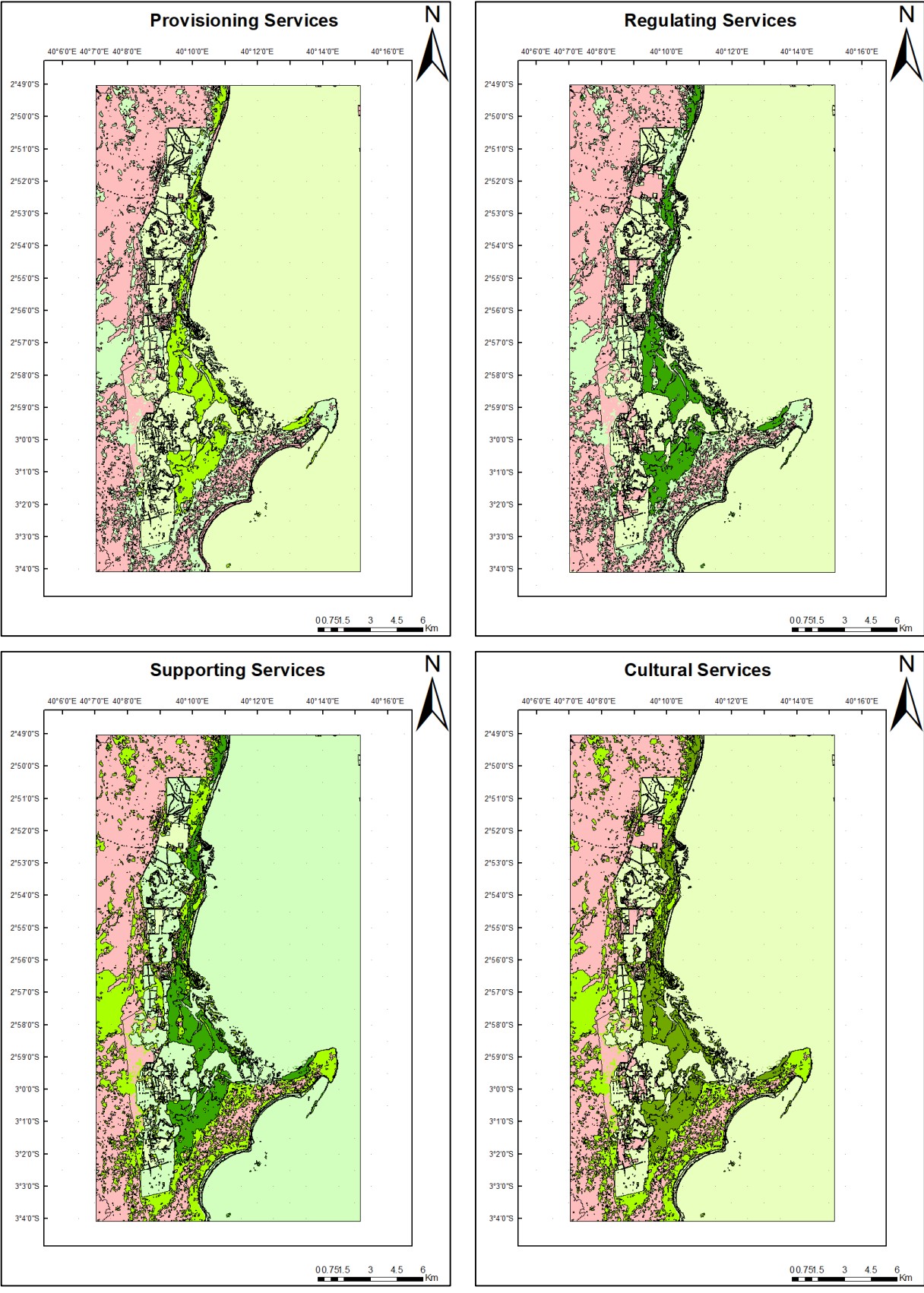

**Figure 6.** *Cont.*

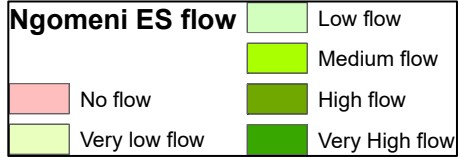

**Figure 6.** The spatial distribution of ES flow for provisioning, regulating, supporting, and cultural services provided in Ngomeni (Ungwana bay, Kenya).

## 4. Discussion

The global and local decline of mangrove forests, leading to the depletion of ES, has made it essential to understand the dynamics and distribution of mangroves to enhance management strategies [27]. One of the adopted strategies is Integrative Natural Resource Management (INRM), the implementation of which requires tools for spatial analysis to understand the spatial relationship between ecosystems and the socioeconomic system [28]. In this regard, characterizing the LULC classes and the ES flow from such a landscape is an essential part of this action [28,29]. More so, human beings, while buffered against environmental changes by culture and technology, fundamentally depend on the flow of ES [8]. This flow is driven by the demand for the service by society being met by the capacity of different landscapes to supply the ES [10]. Therefore, mapping the distribution and dynamics of ES has the potential to aggregate such complex information [30], with the visualization attribute of ES maps being a starting point from which to open conversations, create landscape databases, and incorporate and implement ES into institutions to influence decision making for INRM [10,28].

Identification of the ES supplied by the mangrove ecosystem of Ungwana bay is among the initial stages of an ES assessment study [31]. The 16 ES identified and assessed emphasize the importance of mangroves as a coastal ecosystem. Several studies [12,20,32,33] along the Kenyan coast have identified similar benefits that the coastal community and environment derive from mangroves. Most of these studies highlight fuel and wood products as the most important service derived from the mangrove ecosystem, while in this study, there was a 100% score for sediment trapping. This finding concurs with the environmental conditions of Ungwana bay, which is prone to sedimentation due to salt harvesting activities coupled with poor upland agricultural and river damming activities [20].

The flow of provisioning services—mainly wood products and fuel from mangroves and other vegetation types—displayed close scores. This could be explained by the mangrove species' preference for fuel, timber, and poles on the Kenyan coast [20,33]. For instance, the *Rhizophora mucronata* and *Ceriops tagal* species were reported as more attractive choices due to their good wood quality and resistance to termites (for construction), as well as their high calorific value and good burning characteristics, even in wet conditions (as fuelwood) [20]. However, while this study scored wood products and fuel from other vegetation types as high flow (4.0), the study by Owuor et al. [12] conducted in Mida creek, on the Kenyan coast, showed that harvesting restrictions imposed by the government on some locations had led local communities to rely on other vegetation types, such as casuarina and palm trees, as sources of fuel and wood, hence scoring medium flow (3.0).

A striking but perhaps expected finding was how few respondents associated freshwater and water purification with the mangrove ecosystem. This kind of response could be explained by the scarcity of safe drinking water on the Kenyan coast. A study in 2009 showed that approximately 17.4% of the coastal population relies on water from wells or boreholes and 10.7% on river/stream water [34]. This situation is exacerbated by the environmental impacts of salt mining in the study area, including saltwater intrusion on groundwater and salinization of surface water sources from the dumping of hyper-saline waste from salt industries [35]. More so, on a global scale, freshwater and fisheries have been identified as the two provisioning services whose levels are beyond sustenance at current demands, much less future ones [8].

The importance of the mangroves of Ungwana bay through the very high (5.0) scoring of the flow of regulating services is further emphasized even in the local context. Nature-based solutions offered by the blue carbon ecosystems such as the mangroves, including coastal protection, conservation and restoration, and climate mitigation and adaptation, are key in the achievement of the 1.5 °C goal set in the Paris agreement [1]. The importance of mangroves on the Kenyan coast is recognized through the government showing interest in initiatives such as a sustainable blue economy and involvement in voluntary carbon trading international markets [34].

Further, the outstanding scoring of the sediment trapping as a supporting service, on the one hand, manifests the important role of mangroves' subsurface roots that bind the soil particles together with their aerial roots changing water flow paths and encouraging sediment deposition [29]. On the other hand, the human-induced pressures placed on the mangroves of Ungwana bay include increased sedimentation caused by the diversion and damming of rivers, poor agricultural practices, and forest degradation, with conditions exacerbated by the salt mining activities in the study area [35].

Finally, cultural services, such as other categories of ES, have a close link to human well-being in offering good social relations, health, and security with different intensity levels [8]. However, services in this category, such as cultural heritage, have been reported to be challenging to quantify, while others, such as recreation and tourism, are easier to quantify using monetary terms [31,36]. This could explain why recreation and tourism and education and research were scored highly across several LULC classes in this study. Its notable that cultural heritage service has, over time, been eroding, with their degradation being attributed to either changes in the ecosystem or overall societal change [8].

The ultimate step in this ES assessment was the visualization of the distribution of ES flow. The coupling of the landscape's capacity to supply services and GIS spatial units, by displaying the ES flow distribution, has high potential in landscape management plans [10], such as the National Mangrove Management Plan (NMMP) and sustainable Blue Economy plans of Kenya. Although matrices deliver a good overview of the ES flow by the mangrove ecosystem [30], the visual effect of maps is a powerful tool for communication, problem identification, and spatial explicit prioritization, all important strategies for supporting adaptive management [37]. According to Maes et al. [37], in the analytical framework for mapping and assessment of environmental conditions in Europe, one of the requirements of the set indicators to measure environmental conditions to inform natural resource-based policy is that it needs to be spatially explicit, considering the current spatial distribution of an ecosystem, which is often derived from LULC information. This gives the use of maps in ES assessment studies a lot of potential to inform and influence policy.

A major hindrance to biophysical assessment studies is the inadequate data necessary for ES quantification [10] associated with a high cost of high-resolution commercial satellite imagery and extensive cloud interference in tropical coastal areas [16]. However, the freely accessible and affordable moderate-resolution satellite data used in this study contain enough spatial resolution to be applied in LULC classification studies [9]. To ensure good quality data, its necessary for the user to conduct adequate pre-processing if working in any area with days of extensive cloud cover. Future efforts to improve the quality of satellite data without compromising the quantity of data would contribute greatly to studies such as this.

In addition, there is a wide array of open-source GIS software such as QGIS, ESA-SNAP, and SAGA-GIS that eliminate users' costs, providing similar functionality to commercial software such as ESRI ArcGIS. QGIS and ESA-SNAP, as applied in this study, provide all the needed requirements for satellite data processing and LULC classification, and thus ES mapping. A distinct advantage of these mapping tools is that they are constantly evolving to cater to the advancement of the data properties and are provided together with tutorials, documentation, and an interactive community forum that support users to learn the necessary skills required for their work. Subsequently, the quantification of the flow of mangrove ES using the LULC matrix is probably the most crucial detail of our research. It

relies on the input of the experts' opinions that reveal significant patterns in the capacity of the different LULC classes to provide ES [10]. The input of experts is a form of stakeholder participation that could contribute to the participatory management of mangroves [23].

## 5. Conclusions

The quantitative and qualitative analysis of the study revealed that among the 16 ES supplied by the mangrove ecosystem of Ungwana bay, their flow ranges from very high flow (5.0) for the regulating and supporting services, high flow (4.0) for the cultural services, and medium flow (3.0) for the provisioning services. Six LULC classes were characterized as being present in the study area, acting as geo-biophysical spatial units for the visualization of the spatial distribution of ES flow. In this study, we used cost-effective materials, freely available satellite imagery, and open-source processing software such as ESA-SNAP and QGIS that are reliable and effective, with potential replication to other areas in the West Indian Ocean (WIO).

Most developing countries are highly reliant on natural resources; hence, INRM is key in ensuring sustainable development. Nonetheless, such nations lack the necessary tools, such as biophysical databases on the potential of different landscapes to supply ES. Therefore, in this study, the application of cost-effective and resourceful approaches eliminates the problem of cost while providing important information. Overall, this study approach contains key attributes of sustainable development via stakeholder engagement and the visual effect of maps which enhance communication between researchers and policymakers.

The findings of this study present the potential of using freely accessible satellite data at a local scale with the possibility of replicating and scaling the methodology to other data-scarce regions, contributing, among other things, to the following:

-   The realization of Kenya's sustainable Blue Economy (under the Vision 2030 flagship) objectives on mapping, data collection, analysis of spatial planning, and ecosystem assessment of natural resources on the Kenyan coast. This is meant to feed the country and regional GIS hub, strengthening databases on land use and environmental change patterns;
-   Addressing the lack of accessible inventory data useful for the mangrove forest conservation and utilization programme, as highlighted in the NMMP of Kenya and UNEP's "A call to action" report.

Future work will map the mangrove cover extent by validating satellite data with in situ data, providing an accurate assessment for this study. This could go beyond our findings and research objectives by covering other areas of research such as land cover change and mangrove species distribution.

**Supplementary Materials:** Supplementary material for the survey of this study can be found here: https://ec.europa.eu/eusurvey/runner/DainaMangroveESsurvey (accessed on 27 January 2023).

**Author Contributions:** Conceptualization, D.M., S.C. and M.A.O.; methodology, analysis, and writing—original draft preparation, D.M.; writing—editing, S.C. and M.A.O.; supervision, S.C. and M.A.O. All authors have read and agreed to the published version of the manuscript.

**Funding:** The first author was supported by a grant funded by the European Commission under the Erasmus Mundus Joint master's degree Program in Water and Coastal Management in the 2018/2019 class (WACOMA; Project num. 586596-EPP-1-2017-1-IT-EPPKA1-JMD-MOB). Sónia Cristina is financed through the FCT under the grant: CEECIND/01635/2017 and would like to acknowledge the financial support of FCT to CIMA through UIDP/00350/2020 and through project LA/P/0069/2020 granted to the Associate laboratory ARNET and Building Capacities of Local Practitioners for the Assessment of the Dynamics of Ecosystems in the Emerging Coastal Towns in the WIO Region (Contract No. WIOMSA/2021/CITIES&COASTS/OB/2021/02).

**Data Availability Statement:** The data presented in this study are from the questionnaires and from publicly available satellite images. The data from questionnaires are available from the corresponding author upon request. The Sentinel-2 satellite images used in this study can be found in an online repository from the European Space Agency Copernicus Open Access Hub (https://scihub.copernicus.eu/dhus/#/home accessed on 15 January 2021).

**Acknowledgments:** We are grateful for the access to free satellite data, the community of Ungwana bay, and the research assistants that contributed to the completion of this study.

**Conflicts of Interest:** The authors declare no conflict of interest.

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
