# Peer review of "Application of Free Satellite Imagery to Map Ecosystem Services in Ungwana Bay, Kenya"

_remotesensing, doi:10.3390/rs15071802_

Round 1

Reviewer 1 Report

Comments and Suggestions for Authors are as follows:

1In the “Introduction” section, it is necessary to include information about relevant research progress and existing main shortcomings, and simplify the research background or significance.  

2The "Results" section should be supplemented with the classification accuracy of LULC.

3In the "Discussion" section, the statements about technological advancement should be removed, as they have already been addressed in the introduction. Additionally, the authors could also discuss the follow-up research plan and outlook in this section.

4In “Conclusions” section, some contents about the significance of this study could be streamlined.

5Noting that the first line needs to be indented by 2 characters, such as P168, and P172.

6For the format of references, please refer to the requirements of remote sensing for typesetting.

Author Response

REPLIES TO COMMENTS OF REVIEWER 1

 B1) SPECIFIC COMMENTS

B1.1) Reviewer’s comments: In the “Introduction” section, it is necessary to include information about relevant research progress and existing main shortcomings and simplify the research background or significance.  

  Author's response: The introduction section was deeply edited incorporating comments from across all the reviewers. Relevant research was included showing how this study builds on it. By incorporating other reviewers' comments the research background was simplified.

B1.2) Reviewer’s comments:、The "Results" section should be supplemented with the classification accuracy of LULC.

 Author's response: The accuracy assessment could not be conducted at the time of the study, but the future work mentioned in the manuscript intends to validate the work within situ data extending into broader studies such as land cover change.

B1.3) Reviewer’s comments: 、In the "Discussion" section, the statements about technological advancement should be removed, as they have already been addressed in the introduction. Additionally, the authors could also discuss the follow-up research plan and outlook in this section.

 Author’s response:  The discussion section was rewritten noting the comments given. Future work on dealing with the problem of cloud interference and extending the research to map the mangrove cover extent by validating satellite data with in-situ data could go beyond our findings and research objectives by covering other research areas such as mangrove species distribution.

B1.4) Reviewer’s comments: In the “Conclusions” section, some contents about the significance of this study could be streamlined.

  Authors’ response: Noted and addressed in the manuscript by ensuring that the statements correlated. See “Track changes”.

B1.5) Reviewer’s comments: Noting that the first line needs to be indented by 2 characters, such as P168, and P172.

  Authors’ response: Noted and edited in the manuscript. See “Track changes”.

B1.6) Reviewer’s comments: For the format of references, please refer to the requirements of remote sensing for typesetting.

 Author's response: Is there a specific reference that is not in the required format?

Reviewer 2 Report

This work aims to assess the flow of ES from the mangrove forests of Ungwana bay, on the North coast of Kenya. It adopts the LULC matrix  to identify ES, characterize the LULC classes present and provide visualization of the flow of ES in Ungwana bay using spatial distribution maps, which is a relevant topic in the field of environmental research.

The manuscript provides a comprehensive overview of the study area and the methods used to obtain and process the satellite imagery. This study highlights the importance of mangroves as a coastal ecosystem and how the visualization of the spatial distribution of ES flow using maps can be useful in informing natural resource managementwhich added to the subject area compared with other published material.

One area that could be improved is the discussion section, which could benefit from a more in-depth analysis of the results and their implications. The authors could also consider adding more detail on the limitations and challenges of using free satellite imagery in mapping ecosystem services and how these limitations could be addressed in future studies.

The conclusions are consistent with the evidence and arguments presented and addressed the main question posed. The references are also appropriate

Generallythis manuscript is well written and scientific sound. I recommend that it be accepted for publication, subject to minor revisions.

Some detailed comments

1) Line 103 “30.881°E, -2.407°S and 40.699°E, -3.280 S”, “-2.407°S” should be “2.407°S” and “-3.280 S” should be “3.280° S”

2) Line 108 “87.5m3.” should be 87.5m3.,

3) Figure 1 is in latitude and longitude coordinate, Figure 2 and 3 is in absolute coordinate, these figures should use same coordinate.

4) Figure 5 and Figure 6, the latitude and longitude coordinate is missing.

Author Response

REPLIES TO COMMENTS OF REVIEWER 2

A2) General comments:

This work aims to assess the flow of ES from the mangrove forests of Ungwana bay, on the North coast of Kenya. It adopts the LULC matrix to identify ES, characterize the LULC classes present and provide visualization of the flow of ES in Ungwana bay using spatial distribution maps, which is a relevant topic in the field of environmental research.

The manuscript provides a comprehensive overview of the study area and the methods used to obtain and process the satellite imagery. This study highlights the importance of mangroves as a coastal ecosystem and how the visualization of the spatial distribution of ES flow using maps can be useful in informing natural resource management, which added to the subject area compared with other published material.

One area that could be improved is the discussion section, which could benefit from a more in-depth analysis of the results and their implications. The authors could also consider adding more detail on the limitations and challenges of using free satellite imagery in mapping ecosystem services and how these limitations could be addressed in future studies.

The conclusions are consistent with the evidence and arguments and addressed the main question. The references are also appropriate.

Generally, this manuscript is well-written and scientifically sound. I recommend that it be accepted for publication, subject to minor revisions.

Author’s Response: Thank you for your remarks and appreciation of the work done. The authors reviewed the discussion section to include the major limitation of using satellite imagery in coastal tropical regions and how future studies can help improve the same.

 B2) SPECIFIC COMMENTS

B2.1) Reviewer’s comments: Line 103 “30.881°E, -2.407°S and 40.699°E, -3.280 S”, “-2.407°S” should be “2.407°S” and “-3.280 S” should be “3.280° S”

 Authors’ response: Edited in the manuscript. See Track Changes.

B2.2) Reviewer’s comments: Line 108 “87.5m3.” should be 87.5m3.,

  Authors’ response: Edited in the manuscript. See Track Changes.

B2.3) Reviewer’s comments: Figure 1 is in latitude and longitude coordinate, Figure 2 and 3 is in absolute coordinate, these figures should use same coordinate.

Authors’ response: Corrections were made on all figures placing them in geodetic coordinates.

B2.4) Reviewer’s comments: Figure 5 and Figure 6, the latitude and longitude coordinate is missing.

Authors’ response: Corrections were made to figures 5 and 6 by adding geodetic coordinates.

Reviewer 3 Report

The selected research topic is quite interesting and has been done appropriately. Some minor points remain to clarify.

LN 58-60:” The degradation of mangroves leads to the release of large amounts of carbon back into the environment due to their high carbon pools leading to global warming.” Is this correct?

LN 109-119: This section presents some of the methodologies. Can we include it in the study area?  

Figure 1: Please check the past research paper and prepare an image caption in scientific form

As you explained, data has been collected using a structured questionnaire. How about the sampling process and size of the sample?

Figure 2,3: Please keep the map elements (Scale, legend..etc.) away from the map area/frame 

Author Response

REPLIES TO COMMENTS OF REVIEWER 3

A3) General comments: The selected research topic is quite interesting and has been done appropriately. Some minor points remain to clarify.

Author’s response: This comment is very much appreciated. Thank you for all the comments that help with the improvement of the manuscript.

 B3) SPECIFIC COMMENTS

B3.1) Reviewer’s comments: LN 58-60:” The degradation of mangroves leads to the release of large amounts of carbon back into the environment due to their high carbon pools leading to global warming.” Is this correct?

Author’s response: Yes, mangroves store large quantities of carbon compared to any other forest type. Their degradation (with no rehabilitation) means no more storage and the wet environment which enable carbon storage then dries out and gradually releases carbon back into the environment.

B3.2) Reviewer’s comments: LN 109-119: This section presents some of the methodologies. Can we include it in the study area?  

Author’s response: This section helps to give a bit of a background of the study area environment from the literature. This study did not look at the mangrove species or their properties.

B3.3) Reviewer’s comments: Figure 1: Please check the past research paper and prepare an image caption in scientific form.

Author’s response: The image caption was revised as suggested as “Figure 1. The geographic location of the study area and area of interest on the North coast of Kenya (Landsat-8 image courtesy of the U.S. Geological Survey).”

B3.4) Reviewer’s comments: As you explained, data has been collected using a structured questionnaire. How about the sampling process and size of the sample?

Author’s response: As stated in the methods section, 2 main groups were targeted. A sample size of 100 and used random sampling with the questionnaire distributed evenly across.  A statement was added in the manuscript clarifying this.

 B3.5) Reviewer’s comments: Figure 2,3: Please keep the map elements (Scale, legend..etc.) away from the map area/frame

Authors’ response: Corrections were made to figures 2 and 3 by having uniform geodetic coordinates on all figures in the manuscript and having map elements out of the map frame.

Reviewer 4 Report

Manuscript ID: remotesensing-2215484

Title: Application of free Satellite Imagery to Map Ecosystem Services in Ungwana Bay, Kenya

Submitted to section: Forest Remote Sensing

I think that the ms is interesting because is a valuable method to use satellite data and social data for the management of coastal resources;  in this case Mangrove forests.

Authors should pay more attention and rewrite  Material and methods and Results parts because their great work should be replicated by scientists in other regions.

I found a lack of data (perhaps in supplementary data ?) about the number of interviews and data for the two sites studied. However, I lost myself in Results. Sometimes mangroves, LULC and ES data are presented and commented for the two regions together, other times only for one regions.

Also the statistical approach is not clear if not supported by tables where quantitative  data are reported for each parameter.

Some suggestions are added in the ms.

I think that the ms could be published only after an accurate revision of the text  as explained above.

Author Response

REPLIES TO COMMENTS OF REVIEWER 4

A4) General comments

A.1) I think that the ms is interesting because it is a valuable method to use satellite data and social data for the management of coastal resources; in this case Mangrove forests. I think that the ms could be published only after an accurate revision of the text as explained above

Author’s response: This comment is very much appreciated.

B4) SPECIFIC COMMENTS

B4.1) Reviewer’s comments: Some suggestions are added in the ms.

Author’s response: All the grammatical and editing suggestions made were incorporated into the manuscript with “Track changes” and explanations made on some of the comments.

B4.2) Reviewer’s comments: Authors should pay more attention and rewrite the material, methods, and results parts because scientists in other regions should replicate their great work.

Author's response: We have gone through the methods and results sections again and made a few changes to allow for clarity to the ready. See Track changes

B4.3) Reviewer’s comments: I found a lack of data (perhaps in supplementary data?) about the number of interviews and data for the two sites studied. However, I lost myself in the Results. Sometimes mangroves, LULC and ES data are presented and commented on for the two regions together, other times only for one region.

Author's response: The comment on the lack of data was addressed in the materials and methods of the number of questionnaires disseminated and the number used in the analysis in results. Concerning the location of the study similar to the first comment, the authors went through the manuscript to make sure it was clearly stated. But as noted in the study area section the study area is essentially one location with a similar environmental and demographic background. The only reason for dividing it into 2 subsets shown in figure 1 was to work around the problem of satellite imagery cloud interference.

B4.4) Reviewer’s comments: Also the statistical approach is not clear if not supported by tables where quantitative data are reported for each parameter.

Author’s response: In each statistical case we looked at whether there was statistical significance on each ecosystem service and the class in which it was supplied. We mostly did data cleaning in excel and then one-way ANOVA using R.

Round 2

Reviewer 1 Report

The manuscript's quality has been improved due to revisions made in response to the reviewer's comments.

Author Response

Reviewer comment: The manuscript's quality has been improved due to revisions made in response to the reviewer's comments

Author reply: Your input is much appreciated. The minor spell checks have been addressed.

Reviewer 4 Report

The ms has been improved. I suggest to mention in Tables and Figures the location of the studied sites (Ungwana Bay, Kenya).

Author Response

Reviewer comment: The ms has been improved. I suggest mentioning in Tables and Figures the location of the studied sites (Ungwana Bay, Kenya)

Author reply: Your input is appreciated. The suggestion has been taken into account in the manuscript. See Track changes.